# Self-organization of a doubly asynchronous irregular network state for spikes and bursts

Filip Vercruysse[1,2]*, Richard Naud[3,4], Henning Sprekeler[1,2]*

**1** Department for Electrical Engineering and Computer Science, Technische Universität Berlin, Berlin, Germany, **2** Bernstein Center for Computational Neuroscience, Berlin, Germany, **3** Department of Physics, University of Ottawa, Ottawa, Canada, **4** uOttawa Brain Mind Institute, Center for Neural Dynamics, Department of Cellular and Molecular Medicine, University of Ottawa, Ottawa, Canada

* filipvercruysse@gmail.com (FV); h.sprekeler@tu-berlin.de (HS)

## Abstract

Cortical pyramidal cells (PCs) have a specialized dendritic mechanism for the generation of bursts, suggesting that these events play a special role in cortical information processing. *In vivo*, bursts occur at a low, but consistent rate. Theory suggests that this network state increases the amount of information they convey. However, because burst activity relies on a threshold mechanism, it is rather sensitive to dendritic input levels. In spiking network models, network states in which bursts occur rarely are therefore typically not robust, but require fine-tuning. Here, we show that this issue can be solved by a homeostatic inhibitory plasticity rule in dendrite-targeting interneurons that is consistent with experimental data. The suggested learning rule can be combined with other forms of inhibitory plasticity to self-organize a network state in which both spikes and bursts occur asynchronously and irregularly at low rate. Finally, we show that this network state creates the network conditions for a recently suggested multiplexed code and thereby indeed increases the amount of information encoded in bursts.

**Data Availability Statement:** All relevant data are within the paper and its Supporting information files. In addition, source code is available at https://

## Author summary

The language of the brain consists of sequences of action potentials. These sequences often contain bursts, short "words" consisting of a few action potentials in rapid succession. Bursts appear to play a special role in the brain. They indicate whether a stimulus was perceived or missed, and they are very effective drivers of synaptic plasticity, the neural substrate of learning. Bursts occur rarely, but consistently, a condition that is thought to maximize the amount of information they can communicate. In our article, we argue that this condition is far from self-evident, but requires very special circumstances. We show that these circumstances can be reached by homeostatic inhibitory plasticity in certain inhibitory neuron types. This may sound complicated, but basically works just like a thermostat. When bursts occur too often, inhibition goes up and suppresses them. When they are too rare, inhibition goes down and thereby increases their number. In computer simulations, we show that this simple mechanism can create circumstances akin to those in the brain, and indeed allows bursts to convey information effectively. Whether this

github.com/sprekelerlab/SourceCode_
Vercruysse21.

**Funding:** F.V. was funded for this project by the
Einstein Foundation Berlin (https://www.
einsteinfoundation.de/) via an Einstein Project to H.
S. and Matthew Larkum (grant rant 1-4000015-01-
EF) and by the German Federal Ministry Education
and Research via a Bernstein Award (https://www.
bernstein-network.de/en/the-network/Network-
Partners/BPCN) to H.S. (grant 01GQ1201). The
funders had no role in study design, data collection
and analysis, decision to publish, or preparation of
the manuscript.

**Competing interests:** The authors have declared
that no competing interests exist.

mechanism is indeed used by the brain now remains to be tested by our experimental
colleagues.

## Introduction

Cortical activity consists of irregular sequences of spikes [1], interspersed with bursts of several
action potentials in quick succession [2, 3]. Many cells in the nervous system have specialized
cellular mechanisms for the generation of bursts [4–6], suggesting that they play a special role
in cortical information processing. Burst activity has been associated with a variety of compu-
tational and cognitive functions, including the conscious detection of stimuli [7], the reliable
transmission of information [8] and the induction of synaptic plasticity [9].

Most burst generating mechanisms rely on nonlinear membrane dynamics and are trig-
gered by specific input conditions [4–6]. In pyramidal cells, bursts can be generated by a coin-
cidence of back-propagating actions potentials and synaptic input to the apical dendrite [4].
This associative mechanism could underlie the integration of external sensory signals—reach-
ing the peri-somatic domain—and internally generated signals [10] such as predictions [11,
12] or errors [13–16], which reach the apical dendrite in superficial cortical layers. Based on
the observation that different information streams target different compartments that in turn
generate distinct spike patterns, it was recently suggested that both information streams could
be conveyed simultaneously by means of a multiplexed neural code [17]. Such a multiplexing
could be exploited, e.g., to route feedforward and feedback information in hierarchical net-
works [13, 17, 18].

For bursts to convey information effectively, they need to occur rarely, but consistently
[17]. Neural recordings suggest that this is indeed the case [2, 3, 7]. However, such a condition
is not easily established [1], because bursts are often triggered by nonlinear, threshold-like pro-
cesses. For example, in PCs, bursts can be generated by dendritic calcium spikes, which in turn
arise from the activation of voltage-gated calcium channels [19]. These channels activate a pos-
itive feedback loop upon sufficient dendritic depolarization, effectively introducing a thresh-
old-like condition for the generation of bursts. If dendritic input is too low, bursts will be
absent entirely. If it is too high, bursts will be the predominant form of activity. Both condi-
tions limit the amount of information bursts can transfer. This suggests that neurons should
homeostatically regulate the amount of bursts they emit, by controlling dendritic excitability
or the amount of dendritic input they receive.

A potential candidate for such a homeostatic control of burst activity is dendritic inhibition.
Apical dendrites of cortical PCs receive inhibition from distinct inhibitory interneuron classes
[20, 21], including somatostatin-expressing (SOM) Martinotti cells [22]. SOM interneurons
could be highly effective homeostatic controllers of burst activity, because the dendritic plateau
potentials that underlie burst generation in PCs are very sensitive to inhibition [4, 23, 24]. Yet,
this high sensitivity asks for dendritic inhibition that is finely tuned to the level of dendritic
excitation, i.e., dendritic inhibition should be adaptive. Such a mechanism of preserving a suit-
able level of dendritic inhibition has been theorized to be essential for dendrites to participate
in the coordination of synaptic plasticity [13].

Here, we use a computational network model to show that such a homeostatic control
could be achieved by a simple form of dendritic inhibitory plasticity. We show that this plastic-
ity can be readily combined with other forms of inhibitory plasticity that control cellular activi-
ty levels overall [25]. In recurrent spiking networks, the combination of these two forms of
inhibitory plasticity can establish a doubly irregular state, in which both spikes and bursts

occur irregularly at a consistent, but low rate. Finally, we show that inhibitory plasticity can self-organize dendritic input levels such that a multiplexing of feedforward and feedback input [17] is more robustly preserved.

## Results

In PCs, bursts occur at a low, but consistent rate [2, 3, 7] and are thought to originate from active dendritic processes [4]. We hypothesized that this is the result of a homeostatic control of burst firing, mediated by plasticity of inhibitory synapses onto apical dendrites. But which inhibitory plasticity rules could achieve such a control and what would be the consequences at the network level? To address these questions, we used a current-based spiking network model consisting of excitatory PCs and inhibitory interneurons. To model the burst mechanism of PCs, PCs were described by a simplified two-compartmental model [17, 26]. In short, the PC model contains a somatic compartment and a dendritic compartment that describes the distal apical dendrite. Both are each modelled by a set of differential equations of the adaptive integrate-and-fire type, with a nonlinearity in the dendritic compartment that allows an active generation of dendritic spikes. The two compartments receive bottom-up and top-down input, respectively (Fig 1A) and communicate by passive and active propagation. This model faithfully predicts spike timing of PCs in response to electrical stimulation [26] and reproduces the qualitative features of burst activity when PCs are injected with somatic and dendritic input Fig 1B [4, 17, 26]. Inhibitory interneurons were described by an integrate-and-fire model.

For clarity, we gradually increase the complexity of the network from an uncoupled population of PCs to a feedforward network with inhibition and, finally, a recurrent network with two interneuron classes, representing dendrite-targeting SOM interneurons and soma-targeting parvalbumin-positive (PV) interneurons. The parameters of the interneuron model were adjusted to reflect the properties of these cell classes, specifically the presence and absence of spike-frequency adaptation in SOM and PV neurons, respectively (see Methods).

### Controlling the burst activity of L5 pyramidal cells requires fine-tuning of the excitatory input and noise levels

The computational role of PCs as bursting units depends on how dendritic and somatic inputs are translated into a spike and burst response. Due to the non-linear nature of the burst generation mechanism, we expected that the dynamic range of burst activity is limited when bursts are driven by dendritic excitation alone [1]. The difference between no burst activity and a situation where every somatic spike is dendritically amplified to a burst [4] should be brought about by small differences in dendritic input. We checked this intuition in a population of model PCs by injecting current into both the soma and the dendrite. Spikes were generated by a noisy background input to the somatic compartment, with firing rates that mimic sensory driven activity [7]. The conversion of somatic spikes into bursts was driven by noisy excitatory current to the dendritic compartment, for which we systematically varied the mean and the noise level.

We found that in the absence of noise on the dendritic input, an increase of the mean input to the dendrite leads to a rapid transition from an absence of bursts to a saturated level of burst activity (Fig 1C, light blue trace). The average population burst rate as a function of the dendritic input currents shows a step-like transition, at around 175 pA for our parameter settings. The majority of spikes appear as single spikes (singlets) below this threshold (Fig 1D, no noise condition). Above, all spikes are converted to bursts. The saturation level for bursting activity is determined by the amount of somatic input and potential refractory effects in the dendrite,

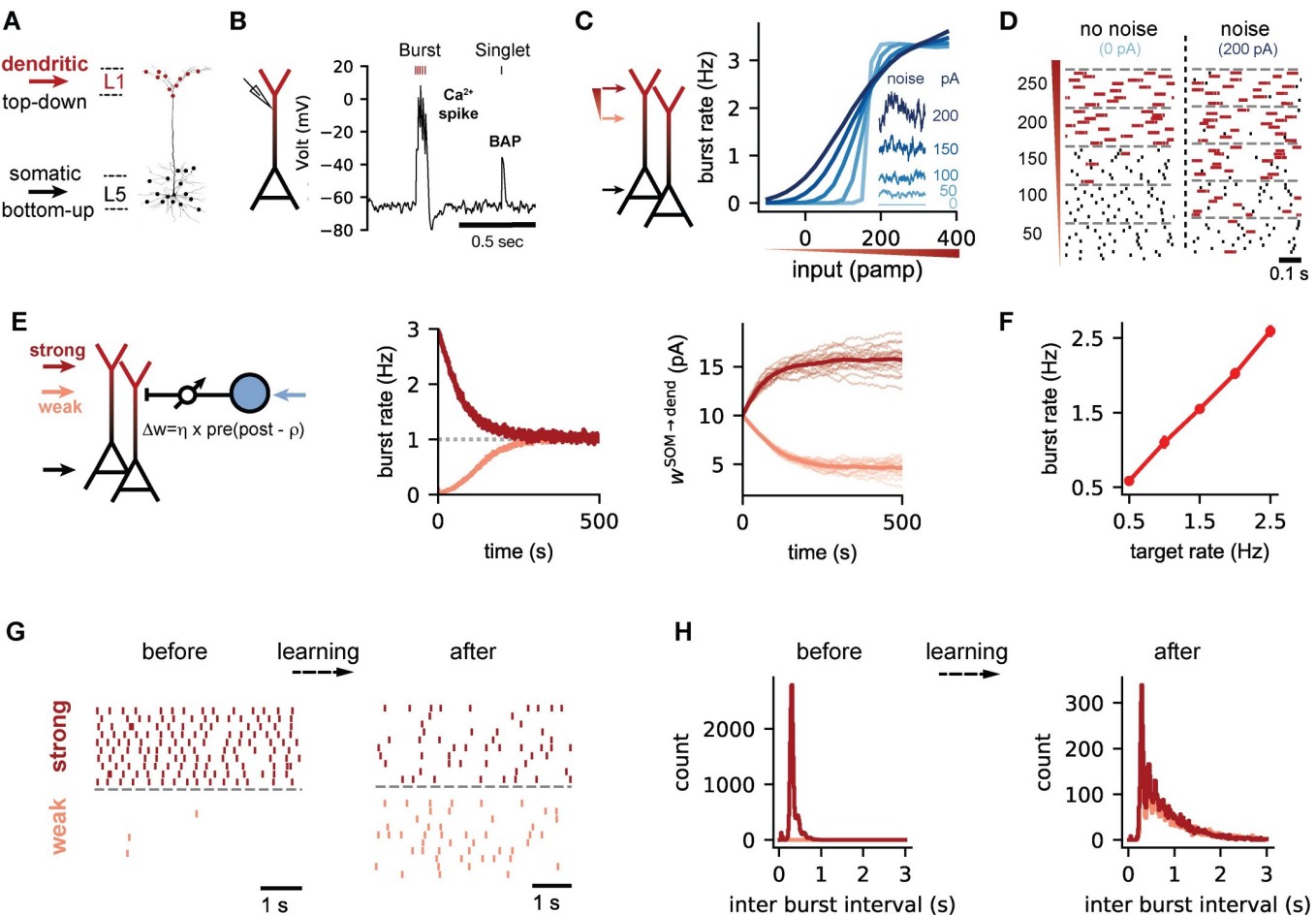

**Fig 1. Control of the burst rate by homeostatic inhibitory plasticity.** (**A**) Anatomy of layer 5 pyramidal neurons (PCs). Sensory bottom-up inputs innervate the perisomatic region (black) while long range top-down connections target the distal dendrites (red). (**B**) Simulated dendritic voltage in a two-compartmental model of PCs. A single somatic spike back-propagates into the distal dendrites. The coincidence of a back-propagating action potential (BAP) with sufficient synaptic input leads to sustained depolarization of the dendrites (calcium spike) and burst activity in the soma [4, 17, 26]. (**C**) PCs are stimulated with varying degrees of dendritic input, characterised by an Ornstein-Uhlenbeck (OU) process. This enables precise control of the mean input (graded red triangle) and noise levels (inset) (See Methods). (**D**) Raster plots illustrating a sharp transition from single spike to burst spikes with increasing dendritic inputs without noise. Noise leads to a more graded transition. Bursts are color coded in red. (**E**) Network configuration with distal dendrites of PCs under control of inhibitory synaptic inputs from SOMs (blue circle). Bursts are activated by weak (pink, $I_i^d = 250$ pA) or strong (red, $I_i^d = 650$ pA) dendritic input with moderate noise levels ($\sigma^d = 100$ pA). The somatic input is the same for both dendritic inputs ($I_i^s = 500$ pA, $\sigma^s = 100$ pA). The strength of the inhibitory connections $w^{SOM \rightarrow dend}$ is plastic (arrow) and modified according to our homeostatic plasticity rule (Eq 1). The burst target rate (dashed line) was set to 1 Hz. (**F**) The burst rate after learning the inhibitory weights for different target burst rates. (**G**) Representative raster plots of the burst activity for weak and strong inputs, before and after learning. Each dot is a burst. (**H**) The distribution of the inter-burst intervals (IBI) before and after learning for weak (pink) and strong (red) dendritic inputs.

which are mediated by a slow adaptation current that hyperpolarizes the dendrite after the sustained depolarization of a dendritic spike (see Methods). Hence, in the absence of noise in the dendrite, the non-linear dendritic threshold mechanism indeed limits the dynamic range of PCs as bursting units. Under this conditions, the low, consistent burst rate observed *in vivo* [2, 3, 7] would require a fine-tuning of the input levels.

Noise can broaden the dynamic range of neural information transmission [27]. We therefore stimulated the dendrite with coloured noise with varying mean and variance (see Methods). Indeed, increasing dendritic noise changes the input-output relation between dendritic input and burst rate from all-or-none to a gradual transition (Fig 1C). Dendritic noise hence

allows for a wider dynamic range for the possible burst rates and reduces the need to fine-tune input levels to achieve sparse bursting. A homeostatic control at low burst rates would therefore benefit from large fluctuations on the dendritic input currents. Large input fluctuations arise, e.g., in balanced networks, in which strong excitatory currents are on average cancelled by strong inhibitory currents [28, 29]. Therefore, we next investigated if dendritic inhibition can mediate a control of burst activity and generate the fluctuations characteristic of a dendritic balanced state.

## Homeostatic inhibitory plasticity controls the burst rate of PCs

Neocortical SOM interneurons specifically target the distal tuft of pyramidal neurons [30] and exert a profound influence on dendritic calcium activity and bursting [4, 23]. To investigate if SOM interneurons can control the burst activity of pyramidal cells, we simulated a postsynaptic population of PCs receiving inhibitory input from SOM interneurons to the dendritic compartment (Fig 1E). Spikes in the SOM and PC population were generated by independent background noise, with firing rates that mimic sensory activity [31, 32].

We considered a burst timing-dependent homeostatic plasticity rule to regulate the strength of inhibitory synapses. In effect, synaptic efficacy is potentiated for near-coincident postsynaptic bursts and presynaptic spikes, while every presynaptic spike leads to synaptic depression. This burst-dependent rule is motivated by a previously proposed homeostatic plasticity rule designed to control postsynaptic firing rates [25], but integrates post-synaptic bursts as salient plasticity-inducing events [9, 13]. The learning rule can be summarised as

$$\Delta w = \eta \times pre \times (bursts - \rho_0), \tag{1}$$

where $\eta$ is the learning rate, $pre$ is presynaptic activity, $bursts$ is a postsynaptic trace reflecting recent burst activity, and $\rho_0$ is a target rate for burst activity (see Methods for details). This rule is supported by experimental data insofar as inhibitory synapses from SOM interneurons onto CA1 pyramidal cells undergo potentiation when presynaptic activity is paired with postsynaptic bursts [33].

We find that this learning rule robustly controls the burst rate of the postsynaptic neuron, both for high and for low excitatory input (Fig 1E, middle), by adjusting the synaptic weights of the inhibitory synapses onto the dendrite (Fig 1E, right). Homeostatic control is robust over a range of target rates covering both low burst rates and bursts rates near saturation (Fig 1F). The learning rule also controlled the temporal patterns of bursting. Before learning, burst activity is dense and sparse for strong and weak dendritic inputs, respectively. After learning, the PCs show similar burst raster plots (Fig 1G) and inter-burst interval (IBI) distributions (Fig 1H) for both initial conditions.

Because somatic burst activity may not be easy to sense for inhibitory synaptic connections on the apical dendrite, we wondered whether inhibitory synaptic plasticity in the dendrite could also be controlled by a postsynaptic signal local to the dendrites. Dendritic calcium spikes generate a long-lasting dendritic plateau potential (S1A Fig, red), which drives somatic bursting during BAC firing. Therefore, a thresholded version of the dendritic membrane potential provides a local estimate of the occurrence of a burst. Using this proxy for burst activity in the homeostatic inhibitory learning rule also leads to robust control of the burst rate (S1 Fig), suggesting that homeostatic burst control could be achieved by a simple, biologically plausible mechanism. In the following, however, we will continue to use the burst-based implementation of the plasticity rule Eq 1, because it allows for the interpretation of the target rate $\rho_0$ as a burst rate.

Note that the dependence of the learning rule on presynaptic activity allows a stimulus-specific form of homeostatic control if the inhibitory interneurons differ in their stimulus tuning [25, 34, 35]. Because the interneurons form a homogenous population in the settings studied here, the presynaptic dependence in the associative term of the learning rule (*pre × bursts*) is not essential and can be dropped (as, e.g., in [36]) without a qualitative change of the results (S2 Fig).

## Simultaneous control of somatic and dendritic activity

Homeostatic inhibitory control of spiking activity has previously been demonstrated in simpler point neuron models without an explicit bursting mechanism [25, 36]. Given the nonlinear interactions between soma and dendrite, we next wondered whether a simultaneous control of bursting activity and overall spiking activity could be achieved by somatic and dendritic inhibition. To this end, we extended the network model by a second class of inhibitory interneurons whose synapses target the somatic compartment of the PCs, akin to PV interneurons [37]. Both PV and SOM populations are modeled as single compartment neurons, driven by external noisy inputs and provide inhibition to the PCs through current-based synapses (see Methods).

To control both somatic and dendritic activity, we have distinct rules for the two inhibitory connections. SOM→dendrite connections are subject to the plasticity rule in Eq 1, while a different spike timing-dependent inhibitory plasticity rule [25] in the PV→somatic connection controls the overall level of activity (Fig 2A). The two learning rules have separate target rates (e.g., a burst rate of 1 Hz and an overall firing rate of 10 Hz) and different learning rates (see Methods for further details).

We find that both the spike rate and the burst rate reach their respective targets (Fig 2B and 2C), but not necessarily in a monotonic fashion. For example, we observed a transient overshoot of burst activity when both firing rate and burst rate were initially too low (Fig 2B, top right). The underlying reason is that firing rate and burst rate are not independent. A decrease in somatic inhibition not only increases the firing rate, but also the burst rate. Hence, a homeostatic control of firing rate can transiently generate an overshoot in burst rate, which is only later corrected by an increase in dendritic inhibition (Fig 2B, top left). The character of this transient effect is likely determined by the relative time scales of plasticity in the two synapse types, which in turn depend on the respective learning rates and the activity in the network.

Homeostatic control is achieved over a range of target values for the firing rate and and the bursts rate (Fig 2C). Conflicts between the two learning rules only arise when the target for the firing rate is too low compared to the target for the burst rate (S3 Fig). This is not surprising, because the firing rate introduces an upper limit for the burst rate. A firing rate of 10 Hz does not allow a burst rate higher than 5 Hz because bursts must by definition contain at least two spikes. A simultaneous control of firing rate and burst rate can also be achieved in a recurrent microcircuit, in which the PV and SOM interneurons receive excitatory input exclusively from PCs (see Fig 3 of the next section). Thus, self-organised inhibition with local learning rules allows a precise control of somatic and dendritic activity in cortical microcircuits, by balancing somatic and dendritic excitation by suitable levels of inhibition.

## A doubly asynchronous irregular state for both spikes and bursts

Asynchronous irregular (AI) activity is a hallmark of recurrent networks in which excitation is balanced by inhibition, and can persist even in the absence of external noise sources [28, 38]. Earlier work has shown that homeostatic inhibitory plasticity can establish such a fluctuation-driven AI state [25]. We therefore hypothesised that the combination of rate and burst

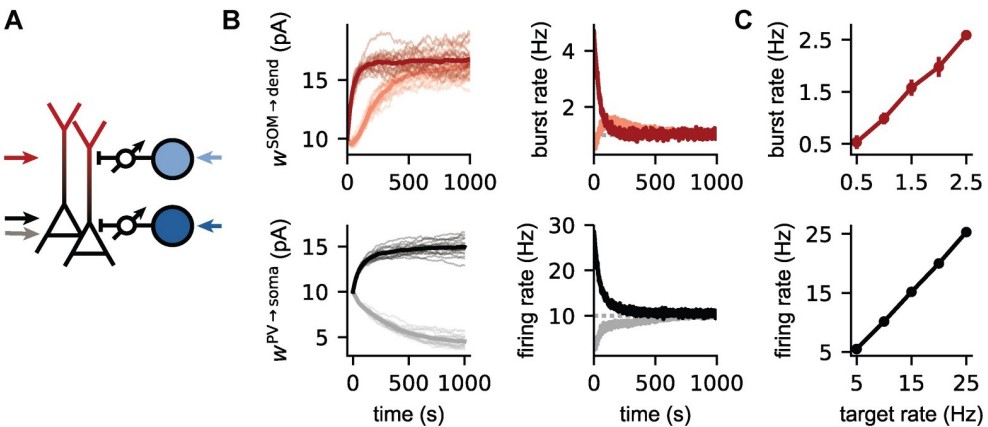

**Fig 2. Simultaneous control of somatic and dendritic activity. (A)** The somatic and dendritic activity of PCs is under control of plastic inhibitory connections from PV (dark blue) and SOM (light blue) interneuron populations. The somatic compartment receives either weak (grey arrow, $I_i^s = 700$ pA) or strong (black arrow, $I_i^s = 1100$ pA) external inputs. The external input to the dendritic compartment is fixed (red arrow, $I_i^d = 650$ pA). The variability of the noisy background input on the external inputs to the PCs is moderate ($\sigma^s = \sigma^d = 100$ pA). **(B)** Evolution of the inhibitory weights, the firing and burst rates of PCs during the learning process for strong (black/dark red) and weak (grey/light red) external somatic input. The target burst and firing rate are respectively 1 and 10 Hz. **(C)** The burst rate (top) and firing rate (bottom) after learning for different target rates for strong somatic ($I_i^s = 1100$ pA) and dendritic input ($I_i^d = 650$ pA). For all conditions, the target firing rate is 10 times larger than the target burst rate.

homeostasis can self-organize a recurrent neural network into a doubly asynchronous irregular state for both spikes and bursts. We tested this hypothesis in a recurrent microcircuit in which all neurons in the circuit are driven by constant, noise-free external excitatory input (Fig 3A). We varied the strength of the input for the somatic and dendritic compartments of the PCs. In addition to plasticity of the inhibitory connections onto the PCs, we applied the homeostatic inhibitory plasticity rule to the inhibitory recurrent connections within the PV population to desynchronise the PV interneurons.

When we initialize the network with small inhibitory weights, the network initially synchronizes strongly at high firing rates (Fig 3B). At this point the absence of inhibition keeps the dendrites of the neurons in a persistently depolarised state, and an identification of bursts is pointless. Over the course of learning, inhibitory plasticity reduces both the firing rate and the burst rate to their respective targets, and the network develops asynchronous irregular activity patterns (Fig 3C). To assess the degree of irregularity of the activity without confounds from the presence of bursts, we studied the statistics not of individual spikes, but of events [17] (see S4B Fig for a comparison of inter-spike interval and inter-event interval statistics). As events, we define individual spikes and the first spike within a burst (Fig 3C, black). Additional spikes within the burst are ignored. We find that both inter-event intervals and the inter-burst intervals are highly variable after learning (Fig 3D; mean CV of the inter-event interval distribution: 1.24; mean CV of inter-burst interval distribution: 0.77) indicating a doubly irregular state.

One hallmark of the fluctuation-driven, inhibition-dominated regime that underlies the AI state in balanced networks is that the mean input current within the population decreases with increasing external drive, while firing rates increase due to an increase in input variance. When we systematically varied the external drive to both the soma and the dendrites of PCs, we find that this is also the case for both compartments (Fig 3E and 3F) in our network model. Note that the external inputs are noise-free, i.e., the variance in the inputs is generated

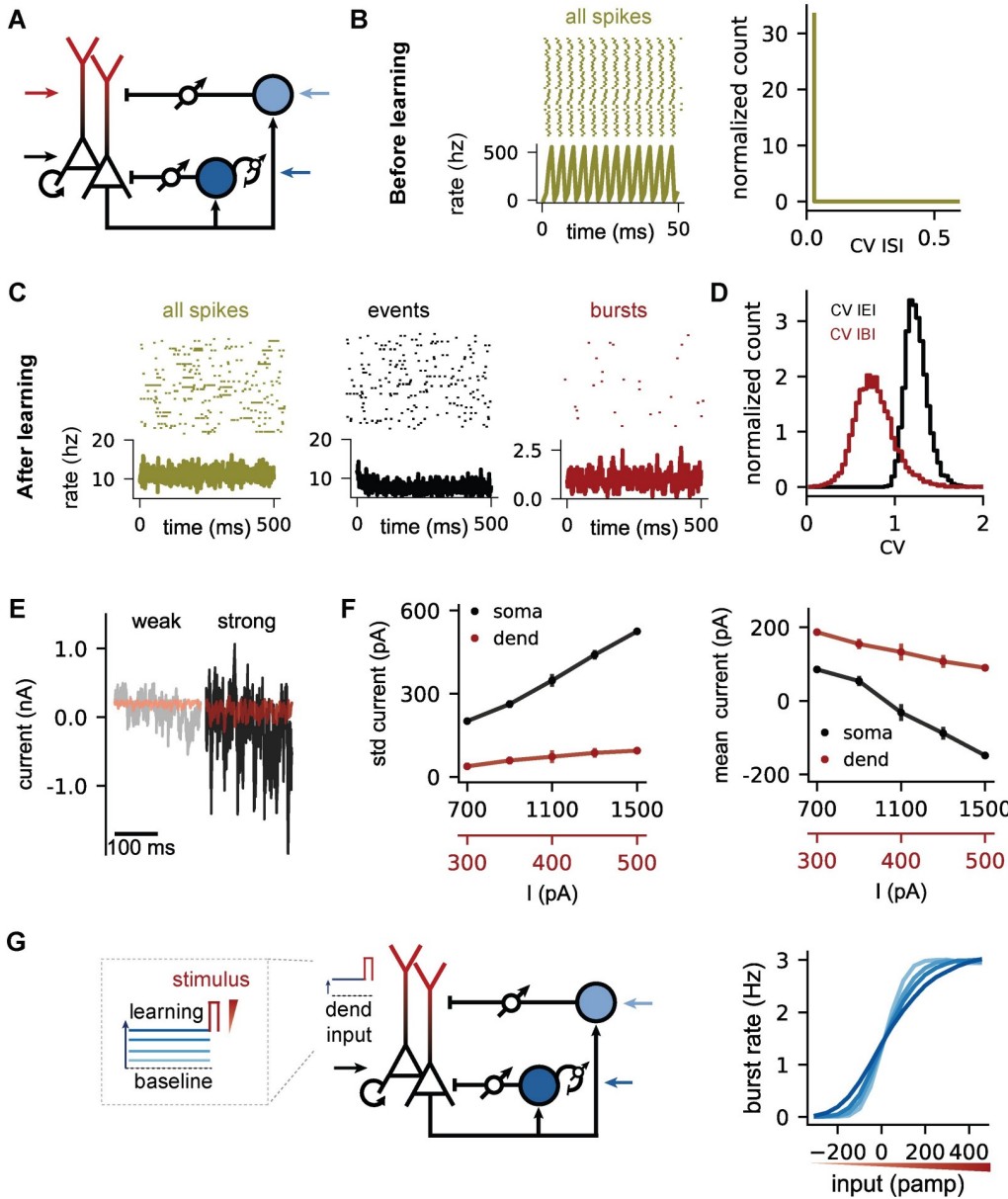

**Fig 3. A doubly asynchronous irregular state for both spikes and bursts. (A)** Schema of the recurrent circuit structure. All inhibitory connections are plastic (target firing rates = 10 Hz, target burst rate = 1 Hz). For all panels, the external input to soma, dendrites and PVs ($I_i^{PV} = 200$ pA) is noise-free ($\sigma^s = \sigma^d = \sigma^{PV} = 0$ pA). SOM interneurons do not receive external input. **(B)** Left: Spiking pattern of PCs before learning of the inhibitory weights and time-varying population rate. Strong constant somatic ($I_i^s = 1500$ pA) and dendritic ($I_i^d = 500$ pA) external input causes PCs to fire synchronously at high rates. No clearly separated bursts are detectable. Right: Regular firing indicated by the distribution of the coefficient of variation (CV) of the inter-spike interval. **(C)** Raster plots and population rates after learning the inhibitory weights. Events (black) and bursts (red) are isolated from all spikes (dark yellow) to illustrate activity associated with somatic (events) and dendritic (bursts) inputs (see Methods). **(D)** The firing and burst pattern is irregular as indicated by the distribution of the coefficient of variation (CV) of the inter-burst (red) and inter-event (black) intervals. **(E)** Net (exc + inh) input current to the soma (black traces) and dendrites (red traces). Weak ($I_i^s = 700$ pA, $I_i^d = 300$ pA) and strong ($I_i^s = 1500$ pA, $I_i^d = 500$ pA) external input leads to small and large input fluctuations on the net input current, respectively. **(F)** Standard deviation (left) and mean (right) net input currents for soma (black) and dendrites (red) when increasing the external somatic (black X-axis) and dendritic (red X-axis) inputs simultaneously. **(G)** Left: Microcircuit and stimulation paradigm. During the learning phase, the inhibitory weights change until the somatic (10 Hz) and dendritic target (1.5 Hz) is reached for constant dendritic inputs ranging from weak to strong (blue $I_i^d = 250$-350-450-650 pA). Right: Burst rate in response to a transient input stimulus (red) after learning, as a function of the strength of the stimulus (red triangle). Somatic input (black arrow, $I_i^s = 1000$ pA) is the same for all dendritic input conditions.

intrinsically by the balance of excitation and inhibition, as for networks with simpler neuron models [28]. Functionally, this internally generated noise has the effect of smoothing out the input-output function of the dendrites, such that transient dendritic inputs are represented in the burst rate in a graded rather than an all-or-none fashion (Fig 3G, cf. Fig 1B). In summary, homeostatic inhibitory plasticity in somatic, dendritic and inter-interneuron connections can establish a doubly asynchronous irregular state in the network, in which both spikes and bursts occur irregularly, by means of internally generated noise.

### Inhibitory plasticity enables a multiplexed spike-burst code

While dendritic inputs to PCs are usually interpreted as "gain modulators" of PC responses [39–41], spikes and bursts could also be used in a multiplexed ensemble code that allows to decode both the somatic and the dendritic input to a neuronal population [17]. According to this hypothesis, somatic input to the PCs is represented in the event rate of a population of PCs, while dendritic inputs are represented by the fraction of events that are bursts (burst fraction, BF; Fig 4A). Indeed, the two time-varying input signals to somata and dendrites are accurately decoded from the event rate and burst fraction of a population of uncoupled PCs (Fig 4B). However, this code is not robust to changes in input conditions. For the encoding of graded signals, the multiplexed code relies on noise in the input signals that smoothens out the neuronal input-output function and effectively decorrelates the responses of different neurons in the population. In other words, the population is artificially maintained in a fluctuation-driven regime by the addition of external noise. In line with this intuition, the decoding accuracy for both the somatic and dendritic input degrades when we add a constant baseline input to the dendrites (Fig 4C–4E), shifting the neurons away from the fluctuation-driven and towards a mean-driven regime.

Given that the two forms of plasticity tend to establish a fluctuation-driven regime, we hypothesised that they could provide a basis for a self-organization of a multiplexed event-burst code. We first tested in an uncoupled PC population, if somatic and dendritic homeostatic plasticity can compensate for the input level-dependent disruption of the multiplexed code. Indeed, the code is recovered over the course of learning for a broad range of input levels (S5 Fig). Finally, we checked if the code can also be enabled in a fully recurrent setting without external noise (cf. Fig 3). Again, we find that inhibitory plasticity can compensate for inappropriate baseline currents (Fig 4F–4I). Over the course of learning, the plasticity recovers a fluctuation-driven state (Fig 4G), and the resulting irregular asynchronous activity enables the multiplexed code (Fig 4H and 4I).

In summary, homeostatic plasticity of inhibitory synapses can put neural circuits with complex PC dynamics and two different interneuron classes into a fluctuation-driven regime, which not only establishes a doubly balanced state, but also stabilizes a multiplexed neural code for bottom-up and top-down signals.

## Discussion

The question of how pyramidal neurons integrate bottom-up and top-down information streams has received keen interest over the past decades. Here, we addressed the question how a network can self-organize into a dynamical state in which this integration is likely to be most effective. We have shown i) that a simple form of inhibitory plasticity can homeostatically control the burst rate, ii) that it can be readily combined with a homeostasic control of firing rate, iii) that this form of homeostasis can establish a doubly irregular network state for spikes and bursts, and iv) that this state indeed improves the ability of bursts to convey information in a multiplexed neural code [17].

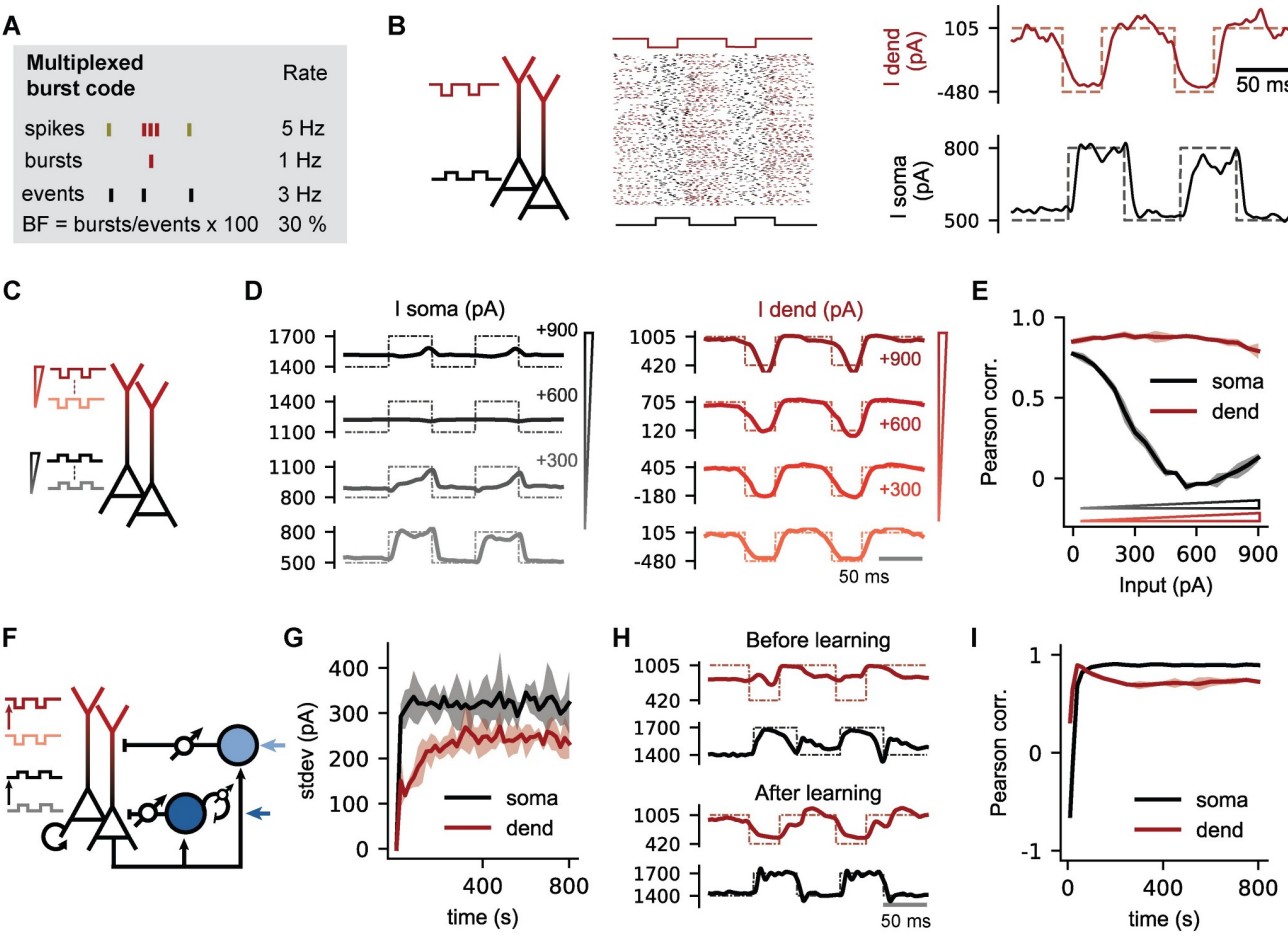

**Fig 4. Inhibitory plasticity self-organises a multiplexed burst code. (A)** Example illustrating a multiplexed burst code in PCs in which somatic and dendritic inputs are represented in the event rate and the burst fraction, respectively [17]. Events are either a burst or single spike, while the burst fraction (BF) is the fraction of events that are bursts. **(B)** Alternating and opposite pulse inputs (dashed lines) are delivered to the somatic and dendritic compartment ($I_i^{s,high} = 800$ pA, $I_i^{s,low} = 500$ pA, $I_i^{d,high} = 105$ pA, $I_i^{d,low} = -480$ pA, $\sigma_i^s = \sigma_i^d = 450$ pA). The pulse inputs can be decoded from the event rate (solid black) and BF (solid red) respectively (see Methods). **(C)** Stimulation paradigm with an increase in background excitation (300, 600 and 900 pA; red triangle = dendrite, black triangle = soma) on which pulse inputs are superimposed. **(D)** Comparison of somatic input/event rate (left) and dendritic input/burst probability (right). Event rate and BF were rescaled using a linear decoder. Dashed lines represent the true inputs. The multiplexed code deteriorates when dendritic and somatic background currents are increased relative to Fig 4B. The values on the y-axis are external input strengths in pA. **(E)** Quality of the multiplexed burst code for increased background excitation, measured by the Pearson correlation coefficient between the two input currents and event rate (black) and burst fraction (red), respectively (see Methods). **(F)** Inhibitory plasticity restores the multiplexed burst code in a biological microcircuit without the need for fine-tuning the background input or noise levels. The microcircuit is similar to Fig 3, with constant external inputs, recurrent connections and plasticity on all inhibitory connections. Background excitation to both somatic and dendritic compartments was increased with 900 pA, where event rate (BF) is not informative of somatic (dendritic) input pulses. **(G)** The learning process increases the standard deviation of the net dendritic (red) and somatic (black) input currents. **(H)** Decoded inputs from the event rate (black) and BF (red) before and after learning, as in D. **(I)** Pearson correlation between actual and decoded inputs to quantify the quality of the multiplexed burst code over the course of learning (see Methods).

## Functional benefits of homeostatic burst control

Given the broad range of potential dendritic computations [42], a homeostatic control of dendritic and/or burst activity could serve a variety of functions. Sparse bursting maximizes information in multiplexed neural codes. Such multiplexed codes could in turn allow a bidirectional propagation of signals in cortical hierarchies [17], e.g., a backpropagation of internal predictions [11, 12, 43] or error signals [13, 18] to lower layers in the hierarchy. In line

with the notion that bursts could represent error signals, they are very effective drivers of synaptic plasticity [9], suggesting that learning can be regulated or at least influenced by inputs to the upper cortical layers. Notably, error-driven learning is substantially more effective when the error signals are graded rather than all-or-none. Therefore, the suggested homeostatic control of burst activity with the accompanying response linearization may be beneficial to create suitable conditions for graded learning signals [13]. Finally, if plasticity is primarily triggered by bursts, a homeostatic control of burst activity could be interpreted as a form of "meta-plasticity" that controls how often plasticity is triggered in a given neuron or circuit. A similar argument can be made for the idea of coincidence detection by somato-dendritic integration [10]. Homeostatic inhibitory plasticity in the dendrite could serve to set a (potentially soft) threshold above which dendritic input is deemed sufficiently high to trigger the coincidence detection machinery.

A different argument for homeostatic inhibitory plasticity is the establishment of a balanced state [44, 45], in which excitation and inhibition cancel out on average [28, 38]. While the energy expenditure of balanced states is a frequent target of mockery (but see [46]), the underlying idea of inhibitory negative feedback loops has the advantage of smoothing out threshold-like processes and thereby broadening the dynamic range for information transmission (Fig 1, [27]). In line with this idea of response linearization, responses to dendritic stimulation are more graded *in vivo* than *in vitro* [23].

## Specificity of homeostatic control

A frequent question for excitation-inhibition (E/I) balance is that of its spatiotemporal precision, i.e., the question along which dimensions excitation and inhibition are correlated and how tight this correlation is [44]. Originally suggested as a balance on the network level and merely present on average across time and neurons [28], the E/I balance can also be specific in time [29, 47], in stimulus space [48, 49], across neurons [50] or across neuronal compartments [51, 52]. Each of these dimensions of specificity has its correspondence in potential inhibitory learning rules that could establish the respective form of E/I balance. Specificity across neurons requires a dependence of inhibitory plasticity on postsynaptic signals [25, 36]. Specificity in time and stimulus requires a dependence on presynaptic activity [45, 53]. A specificity across compartments—as studied here—requires a dependence on compartment-specific signals. In our simulations, we used bursts as a proxy for dendritic activity of L5 pyramidal neurons, but dendritic membrane potential or currents (S1 Fig; [54]), or local chemical signals could be equally suitable. Earlier models of dendritic (inhibitory) plasticity also exploited local dendritic signals, e.g., to learn dendritic predictions of somatic inputs by comparing compartmental membrane potentials [54, 55]. We included a dependence on presynaptic activity in the dendritic learning rule to leave open the possibility of both a compartment- and input-specific E/I balance in further studies. However, in the situations studied here, the presynaptic interneuron populations are homogeneous, so the presynaptic dependence does not have an impact on the results (Fig 1 versus S2 Fig).

## Experimental support and interaction of homeostatic mechanisms

A key prediction of the model is that inhibitory synapses from SOM interneurons onto PCs should undergo potentiation when the postsynaptic cell bursts too often. This is supported by slice experiments in the hippocampus [33], which showed that a theta burst stimulation protocol—presynaptic activity paired with regularly occurring postsynaptic bursts—induces long-term potentiation in SOM→PC synapses. Notably, the same protocol induces long-term depression in PV→PC synapses. Different interneurons hence display different rules of

synaptic plasticity—also on their excitatory input synapses [56] –, which rely on distinct molecular mechanisms [33, 57].

In our simulations, the effects of different forms of homeostatic plasticity are not necessarily independent. Homeostatic control of the overall firing rate also influenced the rate of bursts (Fig 2), because bursts generated via BAC firing [4] are triggered by somatic spikes and hence depend on the firing rate. Such interactions arise when the sensors (firing rate/burst rate) or the effects (SOM→PC/PV→PC synapses; somatic/dendritic membrane potential) of the homeostatic control laws are correlated, and can generate non-monotonic homeostatic dynamics (Fig 2B) (similar behavior was seen, e.g., by O'Leary et al [58]). This creates potential challenges for an experimental investigation of dendrite-specific forms of homeostasis. The basic prediction of our model is that an over-activation or suppression of dendritic activity should result in specific compensatory changes in inhibitory synapses onto the dendrite. However, it may not be trivial to manipulate dendritic activity without manipulating other aspects of network activity. Depending on the experimental manipulation (e.g., tissue-wide application of TTX [59] vs. application of gabazine or baclofen to the superficial layers [23]) and the observed quantities (burst rate, dendritic calcium signals or morphological features of inhibitory synapses on the dendrite), the observations could differ substantially, if other homeostatic mechanisms occur in parallel. Moreover, it is conceivable that different forms of homeostasis interact to decorrelate their effects. For example, a neuron could react to high dendritic activity by redistributing inhibition from the soma to the dendrite, in order to selectively reduce bursts without affecting overall firing rate. A stimulation protocol based on postsynaptic bursts would then simultaneously potentiate dendritic inhibition and depress somatic inhibition. The observed opposing forms of plasticity in SOM and PV synapses for the same stimulation protocol [33, 57] could therefore be interpreted as a decorrelation of the effects of these two synapse types on firing rate and burst rate.

While inhibitory plasticity is a promising candidate for network homeostasis, it operates on a slow time scale of hours or days and is therefore likely complemented by additional negative feedback loops that operate on faster time scales [60]. Potential mechanisms that could rapidly stop network instabilities in their tracks could be, e.g., short-term synaptic plasticity [61, 62] or presynaptic inhibition [63]. An inclusion of these mechanisms would be interesting, but exceeds the scope of the present study.

### Relation to *in vivo* bursting statistics

*In vivo*, bursts occur rarely, but consistently. In rat somatosensory cortex, the proportion of spikes that occur in bursts is about 15–20% [2]. Similar values were reported by Doron et al [3]. Sanders et al found a slightly higher, but quite variable proportion of about 40% in rat CA1 and CA3 [64]. While firing and burst rates vary substantially across brain regions, the proportion of bursts hence seems to be quite similar across brain regions. Note that a direct quantitative comparison of these numbers is problematic, because not all papers use the same formal criteria for the classification of bursts. In our simulations, we chose the target rates of the plasticity rules such that the proportion of spikes occurring in bursts roughly aligns with the experimental observations. For a firing rate of 10 Hz, a burst rate of 1 Hz and 2–3 spikes per bursts, about 20–30% of the spikes are part of a burst.

### What's wrong or missing in the circuit

The primary focus here was on the self-organisation of a dynamical network state in which bursts occur rarely. Like all models, we navigated a trade-off between model simplicity, clarity of result and biological accuracy, and the circuit we studied is clearly simplified compared to

cortical circuits. For simplicity, we used the same low connection probability among all neuron classes, although the connection probability of interneurons is substantially higher than that of excitatory neurons [65, 66]. We expect that the key results carry over denser connectivity, despite potentially higher input correlations [29].

Several cortical interneuron classes were ignored, including interneuron types that also inhibit the distal dendrite [21, 67]. In principle, those interneurons could provide the suggested homeostatic control of dendritic activity equally well as SOM interneurons. We chose to refer to the dendrite-targeting interneurons as SOM interneurons, because those—specifically deep layer Martinotti cells—receive excitatory drive from the surrounding PC population [23] and are therefore good candidates for the feedback inhibition modelled in Fig 3. Neurogliaform interneurons in layer 1, such as neuron-derived neurotrophic factor-expressing (NDNF) interneurons also target apical PC dendrite, but seem to receive primarily long-range, top-down inputs [21, 68]. Whether two distinct interneuron classes are actually required for a compartment-specific form of feedback inhibition or whether this could be mediated by a single cell class with heterogeneous properties was investigated elsewhere [52]. We also ignored the well-documented connections between SOM and PV neurons [20, 69]. In the presence of stimuli, these connections could mediate a redistribution of inhibition across the two compartments [70], but in the steady-state conditions we studied here, they would likely not change the results qualitatively. Additional interneurons that mediate—e.g., a dynamic disinhibition of the dendritic compartment [70–74]—would also become relevant players in the presence of time-varying inputs.

## Outlook

Natural extensions of this work would be the addition of time-varying or stimulus-dependent input, combined with a stimulus tuning of the various cell classes, to study simultaneously the effects of stimulus-specific [25, 34, 35] and compartment-specific [52] E/I balance. To do so, however, we would have to specify a stimulus selectivity for all neuron classes in the network [34, 35] and the resulting rich combinatorics of conditions is beyond the scope of this work.

## Methods

### Network model

We gradually increase the complexity of the network from an uncoupled population of PCs to a feedforward network with inhibition and, finally, a recurrent network with two interneuron classes, representing dendrite-targeting SOM interneurons and soma-targeting parvalbumin-positive (PV) interneurons. All neurons are randomly connected. Parameters are provided in Tables 1–3.

**Table 1. Parameter values for the two-compartmental PC model.** Soma and dendrite indicate the somatic and dendritic compartment respectively and $f(x)$ the sigmoid function. Values are from [17].

| soma | | | dendrite | | | $f(x)$ | | |
|---|---|---|---|---|---|---|---|---|
| $\tau^s$ | 16 | ms | $\tau^d$ | 7 | ms | $E_d$ | -38 | mV |
| $C^s$ | 370 | pF | $C^d$ | 170 | pF | $D_m$ | 6 | mV |
| $g^s$ | 1300 | pA | $g^d$ | 1200 | pA | | | |
| $b_w^s$ | -200 | pA | $c^d$ | 2600 | pA | | | |
| $\tau_w^s$ | 100 | ms | $\tau_w^d$ | 30 | ms | | | |
| $E_L$ | -70 | mV | $a_w^d$ | -13 | nS | | | |
| | | | $E_L$ | -70 | mV | | | |

**Table 2. Parameters of the PV and SOM interneuron models.**

| PV | | | SOM | | |
|---|---|---|---|---|---|
| $\tau^{PV}$ | 10 | ms | $\tau^{SOM}$ | 20 | ms |
| $C^{PV}$ | 100 | pF | $C^{SOM}$ | 100 | pF |
| | | | $b_w^{SOM}$ | -150 | pA |
| | | | $\tau_w^{SOM}$ | 100 | ms |

**Table 3. Number of neurons in each populations for the different figures.**

| FIGURE | 1 | 2 | 3 | 4 |
|---|---|---|---|---|
| $N^{PC}$ | 1600 | 1600 | 8000 | 8000 |
| $N^{SOM}$ | 400 | 400 | 2000 | 2000 |
| $N^{PV}$ | - | 400 | 2000 | 2000 |

**PCs.** PCs are simulated as a two-compartmental model akin to the model described by Naud et al [17]. The two compartments represent the soma and distal dendrites, and their interaction captures dendrite-depending bursting.

The membrane potential $V^s$ of the somatic compartment follows generalised leaky integrate-and-fire dynamics with spike-triggered adaptation. The subthreshold dynamics of the $i^{th}$ pyramidal neuron is described by

$$\frac{dV_i^s}{dt} = -\frac{(V_i^s - E_L)}{\tau^s} + \frac{g^s f(V_i^d) + I_i^s + w_i^s}{C^s},\tag{2}$$

$$\frac{dw_i^s}{dt} = -\frac{w_i^s}{\tau_w^s} + b_w^s S_i^s.\tag{3}$$

The dynamics of the somatic membrane potential $V_i^s$ are governed by a leak term that drives an exponential decay to a resting membrane potential $E_L$ with membrane time constant $\tau^s$. When the somatic membrane potential reaches a threshold $V_T$ of -50 mV, it is reset to the reversal potential $E_L$ after an absolute refractory period of 3 ms and a spike is added to the spike train $S_i^s$. The soma is subject to spike-triggered adaptation (Eq 3). Each somatic spike increases an adaptation current $w^s$ by an amount $b_w^s$. Between spikes, the adaptation current $w^s$ decays exponentially with time constant $\tau_w^s$. The soma receives external inputs $I_i^s$ and a current $f(V_i^d)$ from the apical dendrite that depends nonlinearly on the dendritic membrane potential $V_i^d$. The parameter $g^s$ controls the coupling strength of the dendrite to the soma. The impact of all these currents on the somatic membrane potential is scaled by the somatic membrane capacitance $C^s$.

The dendritic compartment is modeled by the following dynamics:

$$\frac{dV_i^d}{dt} = -\frac{(V_i^d - E_L)}{\tau^d} + \frac{g^d f(V_i^d) + c^d K(t - \hat{t}_i^s) + I_i^d + w_i^d}{C^d},\tag{4}$$

$$\frac{dw_i^d}{dt} = \frac{-w_i^d + a_w^d(V_i^d - E_L)}{\tau_w^d}.\tag{5}$$

The dendritic membrane potential $V_i^d$ decays exponentially to the resting membrane potential $E_L$, with a time constant $\tau^d$. Dendritic calcium events are modeled as a nonlinear current

$f(V_i^d)$, which increases steeply when the dendritic membrane potential approaches a given threshold $E_d$:

$$f(x) = \frac{1}{1 + \exp(-(x - E_d)/D_m)} \ .$$ (6)

The steepness of this threshold is controlled by the parameter $D_m$. The coupling from the somatic to the dendritic compartment by backpropagating action potentials (BAPs) is modelled by a pulse-shaped current in the dendrite. The strength of this current pulse is controlled by the parameter $c^d$ and its shape by a kernel $K$. $K$ is a rectangular kernel of amplitude one, which lasts 2 ms and is delayed by 0.5 ms relative to the time $\hat{t}_i^s$ at which the somatic spike occurred. The dendrite is subject to subthreshold adaptation, which terminates dendritic calcium events unless external currents do so. The dynamics of the dendritic adaptation variable are defined by a strength $a_w^d$ and time constant $\tau_w^d$. Again, all currents to the dendritic compartment are scaled by the dendritic membrane capacitance $C^d$.

For Fig 3, we removed the spike-triggered somatic adaptation $w^s$ (Eq 3) to allow for short interspike intervals (ISI) between somatic spikes. Because this prolongs dendritic calcium spikes and bursts, we increased the value of the dendritic adaptation variable $a_w^d$ from -13 nS to -28 nS to shorten the bursts to realistic numbers of spikes. The choice to remove adaptation for Fig 3 is a classical trade-off between clarity of result and biological accuracy. The original PC model of Naud et al [26] contains somatic adaptation, because this allows a closer fit to neural recordings. Therefore, the biologically more accurate model choice would include adaptation. On the other hand, adaptation suppresses short ISIs and thereby generates more regular firing [75]. As a result, the CV of the inter-event distribution would be consistently below the value 1 that is considered the hallmark of irregular activity. Unfortunately, this could readily be mistaken for an indication that the network is not a fluctuation-driven state. Therefore, we removed the adaptation to make the result on the double AI state as clean as possible. Putting adaptation back in does not drastically change the network state, but the CV of the inter-event distribution is consistently below 1.

**PV and SOM interneurons.** The dynamics of the two interneuron populations are modelled by integrate-and-fire neurons. The subthreshold voltage dynamics $V_i^{PV}$ of the $i^{th}$ PV neuron is described by

$$\frac{dV_i^{PV}}{dt} = -\frac{(V_i^{PV} - E_L)}{\tau^{PV}} + \frac{I_i^{PV}}{C^{PV}} \ ,$$ (7)

with membrane time constant $\tau^{PV}$ and capacitance $C^{PV}$.

In contrast to PVs, SOMs exhibit firing rate adaptation $w_i^{SOM}$ [20], and are therefore described by an adaptive integrate-and-fire model,

$$\frac{dV_i^{SOM}}{dt} = -\frac{(V_i^{SOM} - E_L)}{\tau^{SOM}} + \frac{I_i^{SOM} + w_i^{SOM}}{C^{SOM}} \ ,$$ (8)

$$\frac{dw_i^{SOM}}{dt} = -\frac{w_i^{SOM}}{\tau_w^{SOM}} + b_w^{SOM} S_i^{SOM} \ ,$$ (9)

where $w_i^{SOM}$ increases by $b_w^{SOM}$ in case of a spike $S_i^{SOM}$ and decays otherwise at a rate defined by $\tau_w^{SOM}$. While PVs have a membrane time constant $\tau^{PV}$ of 10 ms, SOMs are modelled with a longer time constant ($\tau^{SOM}$) of 20 ms to be consistent with *in vivo* measurements [76]. The parameters $\tau^{SOM}$ and $C^{SOM}$ are the membrane time constant and capacitance of SOMs, respectively.

## Connectivity

Specific network configurations were used for the different figures. The number of neurons in the network for the main figures of the manuscript are summarized in Table 3. S1 and S2 Figs have the same network configuration as Fig 1, while S3, S4 and S5 Figs have the same network configuration as Figs 2, 3 and 4, respectively. Between the different neuron populations, all neurons are fully connected in Figs 1 and 2 and S5 Fig (connection probability = 1) and with sparse random connectivity in Figs 3 and 4 (connection probability = 0.02). The recurrent connections between PC and PV neurons in Figs 3 and 4 also have sparse random connectivity (connection probability = 0.02). The network diagram depicted in each figure specifies the synaptic connections between the different neuron populations, with arrows indicating excitatory connections and straight arrow heads indicating inhibitory connections. All cells have the same number of incoming connections (homogeneous network), while autapses were excluded from the recurrent inhibitory $PV_i \rightarrow PV_j$ connections. The excitatory connections in Figs 3 and 4 were not plastic and the synaptic weights $w_{ij}$ are fixed for $PC^s_i \rightarrow SOM_j$, $PC^s_i \rightarrow PV_j$ and $PC^s_i \rightarrow PC^s_j$. The strengths of the synaptic weights $w^{s\rightarrow PV}_{ij}$, $w^{s\rightarrow SOM}_{ij}$ and $w^{s\rightarrow s}_{ij}$ are 25 pA for Fig 3, while for Fig 4 the values are 15, 4 and 13 pA respectively. Reducing the strength of the excitatory connection for Fig 4 minimises the mixing of dendritic and somatic input through inhibitory populations, and hence improves a multiplexed burst code. Note that optimized multiplexing to decode both input streams in a recurrent network setting would require additional network elements, i.e. short term plasticity on the excitatory connections and additional inhibitory connections between the PV and SOM population [17]. Optimizing these connections was beyond the scope of this project but was published in Keijser et al [52].

All inhibitory connections were plastic and evolved according to the inhibitory plasticity rules. The inhibitory weights for Figs 1 and 2 and S1–S3 Figs were initialised at 10 pA while for Figs 3 and 4, S4 and S5 Figs inhibitory weights were initialised at 0.1 pA.

## Inhibitory plasticity

Spiking activity of PCs is regulated by an inhibitory plasticity rule described in Vogels et al [25]. Inhibitory synapses are strengthened by coincident pre- and postsynaptic activity within a symmetric coincidence time window of width $\tau_{STDP}$ (= 20 ms). Additionally, every presynaptic spike leads to a reduction of synaptic efficacy. In order to calculate the changes to each $w_{ij}$, a synaptic trace $x_i$ is assigned to each neuron and $x_i$ increases with each spike $x_i \rightarrow x_i + 1$. Otherwise it decays following

$$\tau_{STDP} \frac{dx_i}{dt} = -x_i. \tag{10}$$

The synaptic weight $w_{ij}$ from neuron $j$ to neuron $i$ is updated for every pre- or postsynaptic event such that

$$w_{ij} \rightarrow w_{ij} + \eta(x_i - \alpha) \quad \text{for presynaptic spikes at time } \hat{t}_j \tag{11a}$$

$$w_{ij} \rightarrow w_{ij} + \eta x_j \quad \text{for postsynaptic spikes at time } \hat{t}_i, \tag{11b}$$

where $\eta$ is the learning rate and $\alpha$ the depression factor. The depression factor $\alpha$ mathematically relates to the target firing $\rho_0$ ($\alpha = 2 \times \rho_0 \times \tau_{STDP}$) as derived in Vogels et al [25].

The burst activity of PCs is controlled by an analogous inhibitory plasticity rule as described above. Again, presynaptic activity $j$ is captured by a synaptic trace that increases with each spike. The postsynaptic activity corresponds to burst activity and requires a different implementation. We explored two different strategies, an algorithmic (Figs 1, 2 and 4, S2, S3 and S5 Figs) and a voltage-based strategy (Fig 3, S1 and S4 Figs). The algorithmic rule increases the

postsynaptic trace $i$ for each somatic burst (see below for our classification criteria for bursts), and decays otherwise with the time constant $\tau_{\text{STDP}}$ (= 20 ms), following Eq 10. The synaptic weight $w_{ij}$ from neuron $j$ to neuron $i$ is updated for every pre- or postsynaptic event as in Eq 11. Similar to the inhibitory rule described in [25], the algorithmic rule mathematically relates a target burst rate $\rho_0$ with the target $\alpha$ by $\alpha = 2 \times \rho_0 \times \tau_{\text{STDP}}$. Note that we used different targets for the two inhibitory plasticity rules.

The algorithmic implementation permits updates of the inhibitory weights as an explicit function of the burst rate, but requires somewhat "non-local" somatic information to update dendritic inhibitory weights. We therefore implemented an alternative rule to demonstrate that the burst rate can also be controlled using local dendritic signals. In this implementation, the post-synaptic activity is represented by dendritic calcium spikes. We approximate dendritic calcium spikes by thresholding the voltage of the dendritic compartment using a sigmoid function (Eq 6) with a sharp threshold at -20 mV ($E_d$ = -20 mV and $D_m$ = 0.01 mV). The relationship between the target $\alpha$ and the burst rate is determined empirically by plotting the burst rate as a function of increasing target values (cf. S1 Fig). Target $\alpha$ = 0.03 and 0.045 for Fig 3C–3E and 3G, respectively. Target $\alpha$ = 0.05 for S1B, S1D and S1E Fig $w_{ij}$ is updated every time there is a presynaptic spike or postsynaptic calcium spike.

The spike-time dependent plasticity rule controlling burst activity can be simplified further to a spike timing-independent model (S2 Fig). In this rule, the changes to $w_{ij}$ do not require coincident pre- and postsynaptic activity and update such that

$$w_{ij} \rightarrow w_{ij} + \eta(x_i - \alpha) \tag{12a}$$

where $\eta$ is the learning rate and $x_i$ is a trace representing postsynaptic burst activity. Similar as described above, an algorithmic and a voltage-based strategy can increase the postsynaptic trace $x_i$, which decays otherwise with the time constant $\tau_{\text{STDP}}$, following Eq 10. The algorithmic rule relates the target rate $\rho_0$ for bursts to the depression parameter $\alpha$ by $\alpha = \rho_0 \times \tau_{\text{STDP}}$ and was used in S2 Fig. The synaptic weights are updated with a fixed regular time interval of 50 ms.

The learning rate $\eta^{\text{SOM} \rightarrow \text{d}}$ is 0.1 for Fig 1, S1 and S2 Figs. For Fig 2 and S3 Fig, the learning rates $\eta^{\text{SOM} \rightarrow \text{d}}$ and $\eta^{\text{PV} \rightarrow \text{s}}$ are 0.1 and 0.01, respectively. For S5 Fig the learning rates $\eta^{\text{SOM} \rightarrow \text{d}}$ and $\eta^{\text{PV} \rightarrow \text{s}}$ are 1 and 0.1, respectively. For Fig 3, S4 Fig and 4 the learning rates $\eta^{\text{SOM} \rightarrow \text{d}}$, $\eta^{\text{PV} \rightarrow \text{s}}$ and $\eta^{\text{PV} \rightarrow \text{PV}}$ are 1, 0.1 and 0.05.

The learning rates were varied for several reasons. Firstly, in Fig 2, we chose the learning rates such that the two forms of plasticity occur on similar time scales, for illustration purposes. Secondly, some of the simulations are computationally costly, because they entail the simulation of spiking networks over the long time scales required for plasticity. We therefore maximised the learning rate for all conditions in which we performed extensive parameter sweeps, to reduce the required simulated time. Third, while the steady state should in principle be independent of the learning rate, this is not necessarily true in practise. For example, the recurrent networks in Fig 3 are (deliberately) initialised in a high-activity, highly synchronous state, in which the notion of bursts is useless. To make sure that this state does not generate pathological weight distributions, we used a high learning rate for the soma-targeting inhibitory synapses, such that the overall activity of the network is rapidly reduced.

## Inputs

The input to the neurons is characterised by external constant input $I^{\text{ext}}$, noisy background input $I_i^{\text{bg}}$ and synaptic input $I_i^{\text{syn}}$:

$$I_i = I^{\text{ext}} + I_i^{\text{bg}} + I_i^{\text{syn}}. \tag{13}$$

The noisy background input $I_i^{\text{bg}}$ is modeled as an Ornstein-Uhlenbeck process with mean $\mu$, variance $\sigma^2$ and correlation time $\tau^{\text{OU}}$

$$\frac{d}{dt}I_i^{\text{bg}} = \frac{[\mu - I_i^{\text{bg}}]}{\tau^{\text{OU}}} + \sigma\sqrt{\frac{2}{\tau^{\text{OU}}}}\,\xi_i\,, \tag{14}$$

where $\xi_i$ is Gaussian white noise with $\langle\xi_i\rangle = 0$ and $\langle\xi_i(t)\xi_i(t')\rangle = \delta(t - t')$. For all simulations $\mu$ and $\tau^{\text{OU}}$ were 0 pA and 2 ms respectively. The parameters of the external input $I^{\text{ext}}$ and the standard deviation of the background input $\sigma$ to PCs are specified in the caption of each figure.

The external and background inputs for the inhibitory SOM and PV populations in Figs 1 and 2 and S1–S3 and S5 Figs was chosen so that the firing rates were 10 Hz (Fig 1, S1 and S2 Figs: $I^{\text{ext,SOM}} = 90$ pA, $\sigma^{\text{SOM}} = 400$ pA; Fig 2, S3 and S5 Figs: $I^{\text{ext,SOM}} = 90$ pA, $I^{\text{ext,PV}} = $ -45 pA, $\sigma^{\text{SOM}} = \sigma^{\text{PV}} = 400$ pA). The background input to SOM and PV populations for Fig 3, S4 Fig and 4 was 200 pA ($I^{\text{ext,SOM}} = I^{\text{ext,PV}} = 200$ pA), but the noise was removed ($\sigma^{\text{SOM}} = \sigma^{\text{PV}} = 0$ pA).

A key point of the recurrent network simulations in Figs 3 and 4G was to show that the network can maintain an AI state based on "internally generated noise" and thereby support a multiplexed code. We therefore removed all external noise ($\sigma^{s,d,SOM,PV} = 0$). The amount of external input to the PV interneurons should not be chosen too low, because it indirectly controls the amount of recurrent inhibition among the PV interneurons. The recurrent inhibitory connections among the PV interneurons aims to achieve a given target rate where higher input leads to stronger inhibitory recurrence. However if the overall excitatory input is too small, recurrent inhibition is removed altogether, which in turn increases the risk of PV interneurons to synchronise and thereby generate network oscillations.

The amount of external input to the SOM interneurons is less critical for the behaviour of the network as long as the relative input of recurrent inputs from the PCs is high enough. In contrast to the PV interneurons, SOM interneurons cannot use recurrent inhibitory connections to desynchronise their activity. We omitted those connections because they are rare in cortex [69]. A desynchronization of the SOM interneurons can therefore only be reached by input fluctuations. Because only the internal recurrent excitation fluctuates, but not the external inputs, the recurrence must account for a sufficiently large fraction of the SOM input to prevent network oscillations. Note that this is a somewhat artificial constraint. In the cortex, there would be both noise from the surrounding network activity and inhibition from other interneurons (e.g., VIP neurons [69]).

The total synaptic input $I_i^{\text{syn}}$ is the sum over all synaptic input currents triggered by all presynaptic neurons where the $f$-th presynaptic spike time of neuron $j$ is labeled $\hat{t}_j^{(f)}$:

$$I_i(t) = \sum_j\sum_f w_{ij}\epsilon(t - \hat{t}_j^{(f)})\,. \tag{15}$$

The time course of the synaptic input is modelled as an instantaneous jump followed by an exponential decay with a time constant of $\tau = 5$ ms for excitatory synapses and $\tau = 10$ ms for inhibitory synapses,

$$\epsilon(t) = \mathcal{H}(t)e^{-t/\tau}\,, \tag{16}$$

where $\mathcal{H}(\cdot)$ is the Heaviside step function and $w_{ij}$ the synaptic weight.

## Data analysis

**Neurometric parameters.**  Bursts are defined as a set of spikes where the interspike interval (ISI) is smaller than 16 ms, followed by a period of quiescence before the next burst occurs.

For Fig 3, in addition to the 16 ms ISI requirement, the presence of a dendritic calcium spike was verified to identify bursts. This was necessary because the absence of adaptation in the somatic compartment (Eq 3) can lead to ISIs below 16 ms in the absence of a dendritic calcium spike. See section on the inhibitory plasticity rules (voltage based strategy) for the identification of a dendritic spike. Events are all isolated spikes and the first spike of a burst. Burst and event rate are calculated by summing the bursts and events across the population, respectively. Burst probability is calculated as the ratio of the burst rate over the event rate. The time-dependent rates are smoothed for display by convolving the burst and event rate with a rectangular window. The window-length is 2.5 seconds for Figs 1 and 2, S1 and S2 Figs and 10 ms for Fig 4. The rates for Fig 3 and S4 Fig are not smoothened in order to evaluate fluctuations in the population rate at a temporal resolution in the ms range. So population rates are computed by dividing the total number of spikes in time-bins of 1 ms by the number of PC neurons (8000) and the bin-size of 1 ms.

**Coefficient of variation.**    To characterize the global state of the network we monitored the interspike intervals of individual spike trains. A hallmark of cortical activity is irregular asynchronous network activity and has a coefficient of variation of interspike intervals (ISI CVs) near 1 [38, 77]. ISI CV values close to zero indicate regular spiking patterns, values near 1 indicate irregular spiking. However burst activity confounds the interpretation of CVs since bursts can increase the CV independent of spiking regularity. To interpret the CV independent of burst activity we quantified the regularity of events (IEI CVs). The regularity of bursts is quantified by computing the coefficient of variation of the inter-burst intervals (IBI CVs).

**Multiplexing error.**    In the multiplexed burst code, spikes are separated in bursts and events to recover the input streams that arrive at the somatic and dendritic compartments of PCs (see [17] and Fig 4A). The encoding quality of the dendritic and somatic input signals (see Fig 4E and 4I) is measured by comparing the shape of dendritic input $I(t)^d$ with the shape of burst fraction BF(t) and the shape of somatic input $I(t)^s$ with the shape of event rate ER(t). We performed these comparisons by means of Pearson correlation coefficients (see Fig 4E and 4I). To visually illustrate the similarity of the two signals (Fig 4B, 4D and 4H), we shifted and scaled event rate and burst fraction to make them comparable to the input currents, using a linear regression. More specifically, linear regression minimized the mean squared error loss between the somatic (dendritic) input and the event rate (BF).

**Statistics.**    Data points with error bars show the mean over 3 experiments ± 1 standard deviation. Solid lines and shaded regions show the mean and 95% confidence interval, respectively. The network is randomly initialised for each experiment.

## Simulation details

All simulations were performed using the Brian simulator version 2.2.2.1 [78]. Differential equations were numerically integrated using the Euler integration method with a time step of 0.1 ms. Source code for inhibitory control of network activity is available on github (https://github.com/sprekelerlab/SourceCode_Vercruysse21).

## Supporting information

**S1 Fig. Control of the burst rate by a voltage based homeostatic inhibitory plasticity rule.** **(A)** Network configuration with distal dendrites of PCs under control of inhibitory synaptic inputs from SOMs (blue circle). The inhibitory connections are plastic (arrow) and modified according to a homeostatic plasticity rule where post-synaptic activity is modelled by a filtered version of the dendritic voltage (right, red trace)(Methods). **(B)** Bursts are activated by weak (light red, $I_i^d$ 250 pA) or strong (dark red, $I_i^d$ 650 pA) dendritic input with moderate noise

levels ($\sigma^d$ = 100 pA). The somatic input is the same for both dendritic inputs ($I_i^s = 500$ pA, $\sigma^s$ = 100 pA). The target value was determined empirically (see C) so that the burst rate was 1 Hz (dashed line). **(C)** The burst rate after learning the inhibitory weights for different target values. **(D)** Representative raster plots of the burst activity for weak (light red) and strong (dark red) dendritic inputs, before and after learning. Each dot represents a burst. **(E)** The distribution of the inter-burst intervals (IBI) before and after learning for weak (light red) and strong (dark red) dendritic inputs.
(TIF)

**S2 Fig. Control of the burst rate by spike-timing-independent homeostatic inhibitory plasticity. (A)** Network configuration with distal dendrites of PCs under control of inhibitory synaptic inputs from SOMs (blue circle). Bursts are activated by weak (light red, $I_i^d = 250$ pA) or strong (dark red, $I_i^d = 650$ pA) dendritic input with moderate noise levels ($\sigma^d$ = 100 pA). The somatic input is the same for both dendritic inputs ($I_i^s =$ pA, $\sigma^s$ = 100 pA). The strength of the inhibitory connections $W^{\text{SOM}\rightarrow\text{dend}}$ are plastic (arrow) and modified according to a homeostatic plasticity rule dependent on dendritic post-synaptic activity (Methods). The burst target rate (dashed line) was set to 1 Hz. **(B)** The burst rate after learning the inhibitory weights for different target burst rates. **(C)** Representative raster plots of the burst activity for weak (light red) and strong (dark red) dendritic inputs, before and after learning. Each dot represents a burst. **(D)** The distribution of the inter-burst intervals (IBI) before and after learning for weak (light red) and strong (dark red) dendritic inputs.
(TIF)

**S3 Fig. Simultaneous control of somatic and dendritic activity without and with competition between inhibitory plasticity rules.** The somatic and dendritic activity of PCs is under control of plastic inhibitory connections from PV (dark blue) and SOM (light blue) interneuron populations (see Fig 2). The somatic and dendritic compartments receive strong external inputs with moderate noisy background input. ($I_i^d = 650$ pA, $I_i^s = 1100$ pA, $\sigma^d = \sigma^s = 100$ pA). **(B, C)** No competition (target firing rate = 10 times target burst rate) versus **(D,E)** competition (target firing rate = target burst rate) between the target burst rate and target firing rate. **(B,D)** The burst and firing rate for different burst and firing target rates after learning the inhibitory weights **(C,E)**.
(TIF)

**S4 Fig. A doubly asynchronous irregular state for both spikes and bursts.** The network configuration and stimulus condition are the same as for Fig 3 **(A)** (Top) Representative raster plots of all spikes, events and bursts of 50 neurons after learning the inhibitory weights. (Bottom) Histogram of all spikes, events or bursts of the entire PC population, normalized by the number of neurons (8000) and binsize (1 ms) to have units of rate. **(B)** The distribution of the coefficient of variation of the inter-spike intervals (CV ISI, yellow), inter-event intervals (CV IEI) and inter-burst intervals (CV IBI) after learning the inhibitory weights. **(C)** The distribution of the inter-spike intervals (ISI, yellow), inter-event intervals (IEI) and inter-burst intervals (IBI) after learning the inhibitory weights.
(TIF)

**S5 Fig. Inhibitory plasticity self-organises a multiplexed burst code. (A)** Stimulation paradigm with an increase in background excitation (triangle, red = dendrite, black = soma) on which pulse inputs are superimposed. Similar to Fig 4, alternating and opposite pulse inputs (dashed lines) are delivered to the somatic and dendritic compartment ($I_i^{s,\text{high}} = 800$ pA, $I_i^{s,\text{low}} = 500$ pA, $I_i^{d,\text{high}} = 105$ pA, $I_i^{d,\text{low}} = -480$ pA, $\sigma_i^s = \sigma_i^d = 450$ pA) and the dendritic and

somatic background is increased by 300, 600 and 900 pA, respectively. Plastic inhibitory connections from PV (dark blue) and SOM (light blue) interneuron populations restore the multiplexed burst code without the need for fine-tuning the background input. **(B)** Decoded input currents from the event rate (solid red) and burst probability (solid black) before and after learning (see Methods). Dashed lines represent the actual dendritic and somatic inputs. (TIF)

## Acknowledgments

We thank Joram Keijser for discussions and feedback on the project, and Owen Mackwood for helpful discussions on network implementation and simulations. They, along with Loreen Hertäg, Laura Naumann and Felix Lundt also provided careful proof-reading of the manuscript.

## Author Contributions

**Conceptualization:** Filip Vercruysse, Richard Naud, Henning Sprekeler.

**Formal analysis:** Filip Vercruysse.

**Funding acquisition:** Henning Sprekeler.

**Investigation:** Filip Vercruysse.

**Methodology:** Filip Vercruysse, Henning Sprekeler.

**Project administration:** Henning Sprekeler.

**Software:** Filip Vercruysse.

**Supervision:** Richard Naud, Henning Sprekeler.

**Writing – original draft:** Filip Vercruysse.

**Writing – review & editing:** Henning Sprekeler.

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
