## [Decision Letter · Decision Letter 0]

2 May 2021

Dear Dr. Vercruysse,

Thank you very much for submitting your manuscript "Self-organization of a doubly asynchronous irregular network state for spikes and bursts" for consideration at PLOS Computational Biology.

As with all papers reviewed by the journal, your manuscript was reviewed by members of the editorial board and by several independent reviewers. In light of the reviews (below this email), we would like to invite the resubmission of a significantly-revised version that takes into account the reviewers' comments.

We cannot make any decision about publication until we have seen the revised manuscript and your response to the reviewers' comments. Your revised manuscript is also likely to be sent to reviewers for further evaluation.

Sincerely,

Lyle J. Graham

Deputy Editor

PLOS Computational Biology

Reviewer's Responses to Questions

**Comments to the Authors:**

Reviewer #1: The study of Vercruysse et al. investigates how inhibitory plasticity can regulate the dynamic range of somatic bursts that depend on dendritic activity. The authors show that plastic dendritic inhibition can linearize burst responses (thus increasing the dynamic range), acting in large part independently by the somatic firing. In the network level, plastic dendritic and somatic inhibition result in irregular states that also support graded burst responses and restore the multiplexed burst code.

This study builds upon previous works of the co-authors, is well designed and elegantly illustrates how plastic inhibition can be exploited for the regulation of dendritic-specific burst mechanisms. It is of potential interest to both computational and experimental neuroscience communities focusing on the field of dendrites. The manuscript is well written, and the code for the simulations will become available upon publication. In general, I find this study very interesting and novel. Nevertheless, I have one major concern regarding the results of the authors that I believe restricts their intuitive interpretation.

Major concern:

My major concern is that some parameters of the model that change between results are not sufficiently justified: For example, and of relevance to the results described in lines 224-239, the learning rate of SOM->d is higher than the learning rate of PV->s in Fig. 2,4 and lower in Fig. 3. An intuition/description of the different behaviors when using the respective values will increase their understanding. Another such example is in Fig. 3 (and Fig. 4), whereby the authors chose not to excite SOM with external inputs. What is the reasoning for this, or else, when/why does the feedforward excitation of the SOM interneurons prohibit the generation of asynchronous states/multiplex code?

Minor:

Line 127-129: The authors state that: “we expected that the dynamic range of burst activity is limited when bursts are triggered by excitatory synaptic potentials on the apical dendrite alone”. Do they authors mean burst generation only due to calcium channels in the apical dendrites (without the coincidence of somatic AP), or bursts that do not dependent on e.g. persistent sodium channels at the soma?

Line 217: For ease of reading, please indicate the difference of the two rules (difference in target rates, difference in learning rates).

Fig. S3, panel labeling seems wrong.

Line 474: Please correct to nS (instead of pA).

Fig. S4 in not referenced in the main text.

Line 600: Please delete parenthesis.

Reviewer #2: The authors propose a neural network model with two inhibitory learning rules to achieve a target number of bursts and overall activity. Based on previous inhibitory learning rules they propose a new one based on establishing a target burst rate, performed by SOM-like interneurons. This is then combined with a PV-like interneuron to balance out the overall activity with a standard ISP learning rule. The authors then show that this combination is effective at developing an asynchronous state with a combination of single spikes and bursts and seems to address some of the problems with current multiplexing models. This is an interesting study that adds important contributions to our knowledge of multiplexing and bursting more generally in cortical microcircuits. However, it falls short in some aspects that we highlight below, in particular a better explanation of some elements (see below) and a lack of comparison with burst statistics observed experimentally.

Major points

1. For a general audience it would be good to clarify what is meant by a "threshold-like process" (e.g. abstract and intro), as this is central to the argument. At least in the introduction it would be good to expand on this so as to make it clear what encodes this threshold. This term seems later to be changed to non-linear, it would be good to link them or keep them consistent throughout the paper.

2. It is not clear from the results in Fig. 4 whether this effect is specific to *only* one of the event rates. I.e. does the model after learning have a reduced multiplexing error across input rates or is this only specific to one of the them as shown in G, and there is actually a reduction for the other input values (as in D)? This should be clarified. Why not show the model output for all the cases in D after learning? In addition, in Fig 4E and 4I why is multiplexing accuracy measured this way? Wouldn't it be better to use some form of decoder accuracy?

3. In the abstract it is stated "thereby increasing the information encoded in bursts". This appears to never be quantified. The authors could in principle use their previous measure to quantify this. Having this as a new figure would strengthen their point.

4. A potential problem with this network with bursting is that runaway activity could easily develop before balance is established. This is particularly problematic as ISP learning rules typically need a low learning rate. A solution to this problem has recently been put forward based on developmental STP (Jia et al. BioRxiv 2021). This issue should be discussed.

5. One of the motivations stated for this study (e.g. Intro) is to get closer to in vivo like statistics of bursts, but I failed to see a comparison or at least a detailed discussion on this point.

6. What are the implications for biological plausibility of removing the somatic adaptation to allow for shorter ISI in Fig 3? More importantly, why include the somatic adaptation in the other figures if it is removed in fig 3?

Minor points

1. Fig 3B: You state that “Strong constant somatic (Is i = 1500 pA) and dendritic (Id i = 500 pA) external input drives L5 PCs to fire synchronously at high rates without burst activity” but the plot shows firing that has an ISI below 16ms (~4ms) which suggests that it’s all bursting activity. Maybe this is just a question of rewording?

2. Line 135: distal dendritic compartment?

3. Missing arrow (or some other symbol) onto the soma of the SOM and PV interneurons (Fig. 1 and 2) to indicate the background input.

4. There are other interneurons that target distal dendrites, namely the NDNF interneurons. Is there a reason for assuming that the ones modelled here are SOM?

5. In Fig. 3 it would be useful to also show the CVs for single spikes.

6. It should noted that Sacramento et al. introduced a learning rule for SOMs that is very similar to this one, but without explicit bursting mechanisms.

7. The use of ISP within PVs (Fig. 3) seems to be an interesting point that I haven't seen before. The need for this should be extended and it should be discussed as a potential prediction. In addition why do the PVs receive external input but the SOMs don’t?

8. Fig 3E: You state that “Weak (strong) external input leads to small (large) input fluctuations on the net input current” which is ambiguous because we don’t know if you are saying that you changed the external input current to the somatic and dendritic compartments independently or if both were changed. Also, what are the values for the strong and weak external inputs? Please clarify.

9. Fig 3F: Same Q as 3E, are you varying independently or together?

10. Fig 3G: in the text, you state that “this internally generated noise has the effect of smoothing out the input-output function of the network, such that transient inputs to the network can be represented in a graded rather than an all-or-none fashion” but only show the smoothing for the bursting. Does this happen for the normal spiking as well? Please clarify.

11. Fig 4B: Why are there two high somatic inputs chosen and one is a negative current? Is it a typo and there shouldn’t be two high somatic inputs: the -480pA input is probably the low dendritic input?

12. There is something strange in Eq.17 you state that it should rescale the inputs to match that of the burst fraction (BF) or event rate but for example if the min value of the BF is 0 and the max is 1 then what you get out is just the same input which suggests it wasn't scaled at all.

13. Fig S3: Letters referring to the wrong parts of the figure: No competition should be B-C not A-B and competition should be D-E not C-D.

**Have the authors made all data and (if applicable) computational code underlying the findings in their manuscript fully available?**

Reviewer #1: Yes

Reviewer #2: None

PLOS authors have the option to publish the peer review history of their article (what does this mean?). If published, this will include your full peer review and any attached files.

Reviewer #1: No

Reviewer #2: No
---

## [Decision Letter · Decision Letter 1]

24 Sep 2021

Dear Dr. Vercruysse,

We are pleased to inform you that your manuscript 'Self-organization of a doubly asynchronous irregular network state for spikes and bursts' has been provisionally accepted for publication in PLOS Computational Biology.

Best regards,

Lyle Graham

Deputy Editor

PLOS Computational Biology

Reviewer's Responses to Questions

**Comments to the Authors:**

Reviewer #1: I appreciate the response of the authors, that has addressed all my comments. I believe that this work provides interesting insights on the inhibitory control of asynchronous states and multiplexing and is well suited for publication in PLoS Comput. Biol. The authors will make the code available after publication.

Reviewer #2: We thank the authors for the good job in addressing all our points. We think the paper is now clearer and provides a nice contribution to the field.

**Have the authors made all data and (if applicable) computational code underlying the findings in their manuscript fully available?**

Reviewer #1: Yes

Reviewer #2: Yes

PLOS authors have the option to publish the peer review history of their article (what does this mean?). If published, this will include your full peer review and any attached files.

Reviewer #1: No

Reviewer #2: **Yes: **Rui Ponte Costa (with the help of Heng Zhu and Will Greedy)

---

## [Editor Report · Acceptance letter]

21 Oct 2021

PCOMPBIOL-D-21-00578R1

Self-organization of a doubly asynchronous irregular network state for spikes and bursts

Dear Dr Vercruysse,

I am pleased to inform you that your manuscript has been formally accepted for publication in PLOS Computational Biology. Your manuscript is now with our production department and you will be notified of the publication date in due course.

With kind regards,

Andrea Szabo
